# Improved Generalization Bounds for Deep Neural Networks Using Geometric Functional Analysis

## Abstract

Understanding how a neural network behaves in multiple domains is the key to further its explainability, generalizability, and robustness. In this paper, we prove a novel generalization bound using the fundamental concepts of geometric functional analysis. Specifically, by leveraging the covering number of the training dataset and applying certain geometric inequalities we show that a sharp bound can be obtained. To the best of our knowledge this is the first approach which utilizes covering numbers to estimate such generalization bounds.

## 1 Introduction

The problem of generalization of deep neural networks from the perspective of theoretical analysis has recently received a considerable amount of interest (Neyshabur et al., 2015; Zhang et al., 2017; Dziugaite & Roy, 2017; Neyshabur et al., 2018; Golowich et al., 2018; Nagarajan & Kolter, 2018; Daniely & Granot, 2019). More specifically, most of the state-of-the-art bound are based on spectral norm based generalization bounds and have shown to give tighter and sharper bounds compared to conventional ones leveraging PAC Bayesian theory (Bartlett et al., 2017; Neyshabur et al., 2015; 2018). However, the bounds in such works are quite limited as they only apply in cases where the parameters are drawn from a distribution or when they are represented by fewer bits than required (Nagarajan & Kolter, 2018).

In this paper, we provide a novel framework based on the foundational mathematics of geometric functional analysis to obtain a sharp bound. We show that this analysis has many advantages over the conventional conceptions which solely rely on stochastic assumptions ignoring the geometric structure of the neural networks and to the best of our knowledge this is the first work attempts to foray in this direction. We compare our result with that of Neyshabur et al. to show that our bound is better than the bounds derived using spectral norms.

Geometric functional analysis deals with infinite dimensional vector spaces from a geometric perspective (Holmes, 2012), of its many applications it is widely in the theoretical analysis of wavelet theory (Young, 2012). In fact we believe this is the most suited framework for dealing with deep neural networks, as their parameters are generally huge and any frame work that is finite will not cover all possible models. Specifically, our approach involves using the covering number of a vector space to derive a bound for the generalization error of a given neural network. However, computing the covering number of a high dimensional vector space becomes an intractable problem as the size of the input space becomes sufficiently large. Thus, we make use of the Vitushkin's inequality (Friedland & Yomdin, 2014) to bound the covering numbers using the Lesbesgue measure, which is a tractable quantity. This bound along with functional inequalities forms the groundwork by which we extend it generalization problem of neural networks which also takes into account the geometry of the data input. Another by product of the analysis is that the bound obtained is in terms of the Frobenius norm of the weight matrices rather than its spectral norm, which has shown to scale more rapidly with the size of the matrix (Vershynin, 2011).

We approach the neural network in a rather unorthodox manner by treating it as a recurrent polynomial function which can be approximated to some arbitrary degree (Telgarsky, 2017), as it is the most amicable framework for exploiting its geometric properties. We explain this in detail in

the following sections along with other mathematical preliminaries. This is followed by several intermediate theorems, which are then used to derive our generalization bound.

## 2 RELATED WORK

The concept of generalization bounds were introduced in (Bartlett, 1998) rather indirectly by analyzing the probability of misclassification. (Zhang et al., 2017) showed that uniform convergence over data points is what is required to understand the generalization of networks. Dziugaite & Roy (2017) obtained non vacuous bounds. (Neyshabur et al., 2017) compared the effect of norm based control, sharpness and robustness on generalization. (Neyshabur et al., 2018) used spectral norm to give a bound which incorporated the weights of the neural networks. (Arora et al., 2018) obtained sharp generalization bounds in terms of sample complexity. (Bietti et al., 2019) used Reproducible kernel Hilbert spaces for studying regularization of deep neural network from analytic viewpoint.

## 3 MATHEMATICAL PRELIMINARIES

### 3.1 POLYNOMIAL FRAMEWORK OF NEURAL NETWORKS

Consider the mathematical formulation of Neural Networks, a neural network $\mathcal{N}$ with $l$ layers is a function which takes an input vector $x$ and returns a vector $\mathcal{N}(x)$ such that:

$$\mathcal{N}(x) = f_l(W_l \dots f_2(W_2 f_1(W_1 x)) \dots) \tag{1}$$

where $W_i$ is the weight matrix of network at layer $i^{th}$ layer, $f_i$ is the corresponding non-linear activation function. If $x_l = [x_l^1 \dots x_l^{n_l}]$ is the output of the $l^{th}$ layer with $n_l$ nodes, then:

$$x_l = g_l(W_l x_{l-1}) \tag{2}$$

As mentioned earlier if the activation functions are approximated using polynomials then mathematically there are only polynomials at each node at every layer in the network, so in our analysis we consider the activation functions as polynomials with certain degrees. Let $a_{ij}^{(l-1,l)}$ be the weight of the neural network from the $i^{th}$ node in $l-1^{th}$ layer to the $j^{th}$ node in the $l^{th}$ layer and $P_l$ be the Polynomial approximation of the activation function. Then based on the above mathematical description of neural networks the output at $2^{nd}$ layer with $n_2$ nodes is

$$x_2 = \left[ P_1\left(\sum_{i=1}^{n_1} a_{i1}^{(1,2)} x_1^i\right), P_1\left(\sum_{i=1}^{n_1} a_{i2}^{(1,2)} x_1^i\right), \dots, P_1\left(\sum_{i=1}^{n_1} a_{in_2}^{(1,2)} x_1^i\right) \right] \tag{3}$$

Similarly for the $3^{rd}$ layer, $x_3$ can be obtained as

$$x_3 = \left[ P_2\left(\sum_{j=1}^{n_2} a_{j1}^{(2,3)} P_1\left(\sum_{i=1}^{n_1} a_{i1}^{(1,2)} x_1^i\right)\right), P_2\left(\sum_{j=1}^{n_2} a_{j2}^{(2,3)} P_1\left(\sum_{i=1}^{n_1} a_{i2}^{(1,2)} x_1^i\right)\right), \right.$$
$$\left. \dots, P_2\left(\sum_{j=1}^{n_2} a_{jn_3}^{(2,3)} P_1\left(\sum_{i=1}^{n_1} a_{in_2}^{(1,2)} x_1^i\right)\right) \right] \tag{4}$$

Clearly, by standard properties of polynomials the above tuple is also a polynomial. Now, if we look into the pattern followed by each layer then the general form of the output at any arbitrary $l^{th}$ layer can be obtained as

$$x_l = \left[ P_{l-1} \left( \sum_{j=1}^{n_{l-1}} a_{j1}^{(l-1,l)} P_{l-2} \left( \sum_{i=1}^{n_{l-2}} a_{i1}^{(l-2,l-1)} x_{l-2}^i \right) \right), \right.$$

$$P_{l-1} \left( \sum_{j=1}^{n_{l-1}} a_{j2}^{(l-1,l)} P_{l-2} \left( \sum_{i=1}^{n_{l-2}} a_{i2}^{(l-2,l-1)} x_{l-2}^i \right) \right), \dots,$$

$$\left. P_{l-1} \left( \sum_{j=1}^{n_{l-1}} a_{jn_l}^{(l-1,l)} P_{l-2} \left( \sum_{i=1}^{n_{l-2}} a_{in_{l-1}}^{(l-2,l-1)} x_{l-2}^i \right) \right) \right] \tag{5}$$

By the same logic applied hitherto the above tuple is also a polynomial. The reason for such an unconventional formulation of a neural network as in the above framework is to analyze the degree of the polynomials using some of its basic properties. Regarding degree of polynomials we have the following standard results:

For any $c_1, c_2, \dots c_k \in \mathbb{R}$ and polynomials $P_1, P_2, \dots P_k$

$$deg(c_1 P_1 + \dots + c_k P_k) \leq \sup_{i=\{1,\dots,k\}} |deg(P_i)| \tag{6}$$

where $deg(.)$ is the degree of the polynomial and for any polynomial $P_a$ and $P_b$

$$deg(P_a P_b) = deg(P_a) deg(P_b) \tag{7}$$

By applying equation (7) on the first element of the vector $x_l$ from equation (5) we get

$$deg \left[ P_{l-1} \left( \sum_{j=1}^{n_{l-1}} a_{j1}^{(l-1,l)} P_{l-2} \left( \sum_{i=1}^{n_{l-2}} a_{i1}^{(l-2,l-1)} x_{l-2}^i \right) \right) \right]$$

$$= deg\left(P_{l-1}\right) deg \left( \sum_{j=1}^{n_{l-1}} a_{j1}^{(l-1,l)} P_{l-2} \left( \sum_{i=1}^{n_{l-2}} a_{i1}^{(l-2,l-1)} x_{l-2}^i \right) \right) \tag{8}$$

Similarly, by applying equation (6) on equation (8) we obtain

$$deg\left(P_{l-1}\right) deg \left( \sum_{j=1}^{n_{l-1}} a_{j1}^{(l-1,l)} P_{l-2} \left( \sum_{i=1}^{n_{l-2}} a_{i1}^{(l-2,l-1)} x_{l-2}^i \right) \right) \leq deg(P_{l-1}) |deg(P_{l-2})| \tag{9}$$

By iterating over all the layers in the network and applying equation (9) we get

$$deg(P_l) \leq deg(P_{l-1}) deg(P_{l-2}) \dots deg(P_1) \tag{10}$$

We make use of the above inequality in the next sections in obtaining the generalization bound.

## 3.2 COVERING NUMBER

Covering number of a general topological space counts the number of spherical balls needed to cover the entire space (Munkres, 2000). This is relevant to our present analysis as it gleans the behaviour of a neural network on an unknown set by understanding its behaviour on a known one. Formally, $M(\epsilon, V)$ is the $\epsilon$ covering number of the space $V$, conceptually this gives a measure of generalizability of the neural network on potentially infinite and unknown datapoints when trained on a finite subset (i.e. the training dataset). Thus, any bound on the covering number would imply that we can predict the lower limit of the number of datapoints essential for accurate prediction on unknown sets.

### 3.3 VITUSHKIN'S INEQUALITY

The Vitushkin's inequality (Friedland & Yomdin, 2014; Yomdin, 2011) has been widely used in geometric functional analysis for studying the behaviour of level sets of analytic functions (Kovalenko, 2017). Along with its many applications this inequality can be used to bound the covering number of a metric space $V$ using the lesbesgue measure (Burk, 2011; Bartle, 2014).

Formally, let $P(x, n, d) = P(x_1, x_2, \ldots, x_n)$ be a polynomial of degree $d$ in $n$ variables, $\mathbb{B}^n$ be a ball of unit radius in $n$-dimensions and $V_\rho(P)$ be the set of all polynomials that is bounded by $\rho$, i.e.:

$$V_\rho(P) = \{x \in \mathbb{B}^n : |P(x, n, d)| \leq \rho\} \tag{11}$$

Then according to Vitushkin's inequality:

$$M(\epsilon, V) \leq \sum_{i=0}^{n-1} C_i(n, d) \left(\frac{1}{\epsilon}\right)^i + \mu_n(V) \left(\frac{1}{\epsilon}\right)^n$$
$$\leq M_d(\epsilon) + \mu_n(V) \left(\frac{1}{\epsilon}\right)^n \tag{12}$$

where $\mu_n(V)$ is the $n$-dimensional Lesbesgue measure (Bartle, 2014) of the set and $M_d(\epsilon)$ are variables defined as:

$$C_i(n, d) \triangleq 2^i \binom{n}{i} (d - i)^i$$
$$M_d(\epsilon) \triangleq \sum_{i=0}^{n-1} C_i(n, d) \left(\frac{1}{\epsilon}\right)^i \tag{13}$$

### 3.4 METRIC (N,D)-SPAN

Metric (n,d)-span measures the accuracy of approximation of the covering number with Lesbesgue measure, that is to say it determines how well the covering number can be approximated by knowing the Lesbesgue measure of that set. This transforms the intractable problem of computing the covering number of the set $V$ into a tractable one, this is extremely important as this quantifies the accuracy of computational experiments to that of theoretical expressions. More formally, for any subset $Z \subset \mathbb{B}^n$, we define a constant $\omega_d(Z)$ which is the metric (n,d)-span of set $Z$ denoted by:

$$\omega_d(Z) = \sup_{\epsilon \geq 0} \epsilon^n [M(\epsilon, Z) - M_d(\epsilon)] \tag{14}$$

### 3.5 DIFFERENTIABLE RIGIDITY CONSTANT

The differentiable rigidity constant (Yomdin, 2011) measures the minimum variation of a function which is differentiable, i.e. it is a lower on the variation of a function. Formally, let $f : \mathbb{B}^n \to \mathbb{R}$, $f \in C^{d+1}$ where $d \in \mathbb{N}$ and $Q_l(f) = \max_{x \in \mathbb{B}^n} \sum |f^{(l)}(x)|$. Here $C^k$ is the family of all $k$ times continuously differentiable functions and $\sum |f^l(x)|$ is the sum of absolute values of all the partial derivatives of $f$ of order $l$. Then the $d^{th}$ differentiable rigidity constant is given by

$$\mathcal{RC}_d = \inf_{f \in U_d} Q_{d+1}(f) \tag{15}$$

where $U_d$ is the set of all $C^{d+1}$ smooth functions $f(z)$ on $\mathbb{B}^n$ vanishing on $Z$ with $Q_0(f) = 1$.

### 3.6 REMEZ d-SPAN

The Remez $d$-span (Brudnyi & Ganzburg, 1973; Yomdin, 2011) measures the variation of the function when extended from a smaller set to a bigger set, this instrument allows for an extensive analysis

of neural networks when it transitions from the training to the testing dataset. Formally, for any polynomial $P$, set $Z \subset \mathbb{B}^n$ and $d \in \mathbb{N}$ the Remez $d$-span is the minimal number $K$ that is the ratio of supremum of polynomial on the whole set to that of supremum of polynomial on the set $Z$

$$\mathcal{R}_d(Z) = \min \left\{ K : \left| \frac{\sup_{\mathbb{B}^n} |P|}{\sup_Z |P|} \leq K \right. \right\} \tag{16}$$

### 3.7 BRUDNYI-GANZBURG'S INEQUALITY

The Brudnyi-Ganzburg's inequality (Brudnyi & Ganzburg, 1973; Yomdin, 2011) tells us the variation in performance from a continuous train dataset to a test dataset. However, it is not applicable for discreet datapoints as in the case when deep neural networks and we have use of $\mathcal{R}_d(Z)$ and $\mathcal{RC}_d$ to derive the corresponding bound for the discreet dataset. Let $\mathcal{B} \subset \mathbb{R}^n$ be a convex body and let let $\Omega \subset \mathcal{B}$ be a measurable set. Then for any real polynomial $P(x) = P(x_1, \ldots x_n)$ of degree $d$ we have

$$\sup_{\mathcal{B}} |P| \leq T_d \left( \frac{1 + (1 - \lambda)^{\frac{1}{n}}}{1 - (1 - \lambda)^{\frac{1}{n}}} \right) \sup_{\Omega} |P| \tag{17}$$

Here $\lambda = \frac{\mu_n(\Omega)}{\mu_n(\mathcal{B})}$, with $\mu_n$ being the Lebesgue measure on $\mathbb{R}^n$ and $T_d$ is the Chebychev polynomial of the first kind (Rivlin, 1974).

## 4 THEOREMS

Theorem (1) gives a bound of on the differentiable rigidity constant in terms of the Chebychev polynomial and since this has a closed form solution it is computationally efficient to obtain $R_d(Z)$.

**Theorem 1.** *If $\omega_d(Z) > 0$ and $\mu_n(V_1(P)) \geq \omega_d(Z)$ is finite and it satisfies the inequality*

$$R_d(Z) \leq T_d \left[ \frac{1 + (1 - \omega_d(Z))^{\frac{1}{n}}}{1 - (1 - \omega_d(Z))^{\frac{1}{n}}} \right] \tag{18}$$

*Proof.* Suppose $|P| \leq 1$ on $Z$. Clearly $Z \subset V_1(P)$ as $P$ is bounded in absolute value by 1 on $V_1(P)$. Then by applying Brudnyi-Ganzburg's inequality (17) with $\mathbb{B} = \mathbb{R}^n$ and $\omega = V_1(P)$ we get the above expression. $\square$

**Theorem 2.** *For any real number $0 \leq \omega \leq 1$ the following inequality holds*

$$T_d \left[ \frac{1 + (1 - \omega)^{\frac{1}{n}}}{1 - (1 - \omega)^{\frac{1}{n}}} \right] \leq \frac{1}{2} \left( \frac{4n}{\omega} - 2 \right)^d \tag{19}$$

*Proof.* We use the simplified expression of $T_d(x)$ when $x \geq 1$, then by simple algebraic manipulations we get

$$T_d \left[ \frac{1 + y}{1 - y} \right] = \frac{1}{2} \left( 2 \frac{1 + y}{1 - y} \right)^d \tag{20}$$

By substituting $y = (1 - \omega)^{\frac{1}{n}}$ in equation (20) we obtain

$$T_d \left[ \frac{1 + (1 - \omega)^{\frac{1}{n}}}{1 - (1 - \omega)^{\frac{1}{n}}} \right] = \frac{1}{2} \left( 2 \frac{1 + (1 - \omega)^{\frac{1}{n}}}{1 - (1 - \omega)^{\frac{1}{n}}} \right)^d \tag{21}$$

Now, as $\frac{1}{n} < 1$ and as $\omega \leq 1$, by applying Bernoulli's inequality on $y$ we obtain

$$(1 - \omega)^{\frac{1}{n}} \leq 1 - \frac{\omega}{n} \tag{22}$$

Applying equation (22) in (21) leads to the final result

$$2\frac{1+(1-\omega)^{\frac{1}{n}}}{1-(1-\omega)^{\frac{1}{n}}} \leq \left(\frac{4n}{\omega}-2\right)$$

$$\left(2\frac{1+(1-\omega)^{\frac{1}{n}}}{1-(1-\omega)^{\frac{1}{n}}}\right)^d \leq \left(\frac{4n}{\omega}-2\right)^d \tag{23}$$

$$T_d\left[\frac{1+(1-\omega)^{\frac{1}{n}}}{1-(1-\omega)^{\frac{1}{n}}}\right] \leq \frac{1}{2}\left(\frac{4n}{\omega}-2\right)^d$$

$\square$

Theorem (3) gives a relation between the differentiable rigidity constant and the Remez $d$-span, this enables us to relate our generalization bound with Remez $d$-span.

**Theorem 3.** *For any subset $Z \subset \mathbb{B}^n$ and $d \in \mathbb{N}$*

$$\mathcal{RC}_d(Z) \geq \frac{1}{2R_d(Z)} \tag{24}$$

*Proof.* Let $f : \mathbb{B}^n \to \mathbb{R}$, $f \in C^k$, $\mathcal{P}(n,d)$ is the set of all real polynomials of degree $d$ in $n$ variables and for $d \in \mathbb{N}$ we define $E_d(f)$ as

$$E_d(f) = \min_{P \in \mathcal{P}(n,d)} \max_{x \in \mathbb{B}^n} |f(x) - P(x)| \tag{25}$$

Let $P_d(x)$ be the polynomial of degree $d$ for any fixed $d$ such that the minimum is achieved. Thus, equation (25) can we rewritten as

$$E_d(f) = \max_{x \in \mathbb{B}^n} |f(x) - P_d(x)| \tag{26}$$

By applying the triangle inequality on equation (26) for the subset $Z \in \mathbb{B}^n$ we obtain

$$\max_{x \in Z} |P_d(x)| \leq \max_{x \in Z} |f(x)| + E_d(f) \tag{27}$$

By the definition of Remez $d$-span in equation (16) we get

$$\max_{x \in \mathbb{B}} |P_d(x)| \leq \mathcal{R}_d(Z)[L + E_d(f)] \tag{28}$$

where $L = \max_{x \in Z} |f(x)|$. Again, by applying triangle inequality on $\max_{x \in \mathbb{B}^n} |f(x)|$ we get

$$\max_{x \in \mathbb{B}^n} |f(x)| \leq \max_{x \in \mathbb{B}^n} |f(x) - P_d(x)| + \max_{x \in \mathbb{B}^n} |P_d(x)| \tag{29}$$

By combining equation (28) and equation (29) we get

$$\max_{x \in \mathbb{B}^n} |f(x)| \leq E_d(f) + \mathcal{R}_d(Z)[L + E_d(f)] \tag{30}$$

As the inequality in equation (30) holds for any value of $d$ we therefore get

$$\inf_d \left[\mathcal{R}_d(Z)\left[L + E_d(f)\right]\right] \leq \inf_{d \in \{0,...,k-1\}} \left[\mathcal{R}_d(Z)[L + E_d(f)] + E_d(f)\right] \tag{31}$$

Let $E_d^t(f) = \max_{x \in \mathbb{B}^n} \frac{||f^{d+1}||}{(d+1)!}$ and as $E_d(f) \leq E_d^t(f)$ equation (31) further reduces to

$$\max_{x \in \mathbb{B}^n} |f(x)| \le \inf_{d \in \{0,\ldots,k-1\}} [\mathcal{R}_d(Z)[L + E_d^t(f)] + E_d^t(f)] \tag{32}$$

For a finite set $Z \cup \{x\}$ where $x \in \mathbb{B}^n$ we define the function $f_{Z,x}$ as

$$f_{Z,x} = \begin{cases} 0, & \text{if } x \in Z \\ 1, & \text{if } x \in \mathbb{B}^n - Z \end{cases} \tag{33}$$

Let $\check{f}_{Z,x}$ be the analytic extension of $f_{Z,x}$ to $\mathbb{B}^n$. Then by applying equation (32) we obtain

$$Q_0(\check{f}_{Z,x}) \le \inf_{d \in \{0,\ldots,k-1\}} [\mathcal{R}_d(Z)[L + E_d^t(f)] + E_d^t(f)] \tag{34}$$

If $L = 0$ and $Q_0(\check{f}_{Z,x}) \ge 1$ then

$$1 \le \min_{d \in \{0,\ldots,k-1\}} (\mathcal{R}_d(Z) + 1) Q_{d+1}(\check{f}_{Z,x}) \tag{35}$$

Thus, $Q_{d+1} \ge \frac{1}{\mathcal{R}_d(Z)+1}$ and by the definition of the rigidity constant in equation (15) we get our desired result $\mathcal{RC}_d(Z) \ge \frac{1}{2R_d(Z)}$.

$\square$

Let $\rho$ be the minimum distance between any two points in a set $Z$ with cardinality $k$:

$$\rho = \min_{x_1, x_2 \in Z} ||x_1 - x_2|| \tag{36}$$

And for neural network $\mathcal{N}$ with $n$ input nodes and $d_l$ be the degree of approximation the activation function for the $l_{th}$ then the following theorem holds true

**Theorem 4.** *The product of $k$ and $\rho$ is bounded by the metric $(n, d)$-span of the set $Z$ for any $n$ and $d_l$*

$$\omega_{d_l}(Z) \ge k\rho \tag{37}$$

The proof of theorem (4) has been relegated to the appendix. Using all the above theorems we have derived till now the derive the final theorem describing the generalization bound.

By leveraging the bounds introduced in theorems (1), (2), (3) and (4) we now derive our generalization bound for deep neural networks.

**Theorem 5.** *The loss $l(\mathcal{N})$ incurred by the network $\mathcal{N}$ using the loss function $l$ on the super-set $\mathbb{B}^n$ given the loss in the subset $Z$ is bounded by*

$$\mathbb{E}_{\mathbb{B}^n} [l(\mathcal{N})] \le \mathbb{E}_Z[l(\mathcal{N})] + \mathcal{O} \left[ \prod_{i=1}^l \left( \frac{8n}{k\rho} ||W_i||^2 \right)^{d_i} \right] \tag{38}$$

*Proof.* Let us assume that the input $x$ to the neural network is randomly sampled according to a continuous probability distribution (i.e Bayesian assumption). We first obtain the bound for the randomly generated input and thereby extend it to the deterministic case by using the strong law of large numbers.

$V_\rho(P) = \{x \in \mathbb{B}^n : |P(x)| \le \rho\}$ if $Z \subset V_\rho(P)$ then $M(\epsilon, Z) \le M(\epsilon, V_\rho(P))$ and further using Vitushkin's inequality (12) we get

$$M(\epsilon, Z) \le M(\epsilon, V_\rho(P) \le M_d(\epsilon) + \mu_n(V) \left( \frac{1}{\epsilon} \right)^n \tag{39}$$

So $[M(\epsilon, Z) - M_d(\epsilon)]\epsilon^n \le \mu_n(V)$ for all $\epsilon$ and by taking supremum of $\epsilon$ on both sides we get $\mu_n(V) \ge \omega_d(Z)$. On combining equation (18) from theorem (1) and equation (19) from theorem (2) we get

$$R_d(Z) \leq \frac{1}{2} \left( \frac{4n}{\omega_d(Z)} - 2 \right)^d \tag{40}$$

It is easy to see from equation (40) it follows that $\frac{1}{2} \left( \frac{4n}{\omega_d(Z)} - 2 \right)^d \leq \frac{1}{2} \left( \frac{4n}{\omega_d(Z)} \right)^d$ and by applying equation (37) from theorem (4) we can further simplify the equation (40) as

$$R_d(Z) \leq \frac{1}{2} \left( \frac{4n}{k\rho} \right)^d \tag{41}$$

For any $x \in \mathbb{B}^n$ we define a special function which captures the effect of sample points on the generalization behaviour, mathematically we define it as

$$h_l(x_l) = P_l(x_l) - \sum_{j=1}^{k} P_l(z_j) \psi \left( \frac{2}{\rho}(x_l - z_j) \right) ||f_l \circ W_l||_F \tag{42}$$

where $\psi$ is a $C^\infty$ function on $\mathbb{B}^n$ such that $\psi(0) = 1$ and $\psi$ vanishes on the boundary on $B^n$, $x_l$ is the output $l^{th}$ layer embedding for the input $x$ and $\{z_0, \ldots z_k\} \in Z$.

As each function $\psi_j = \psi \left( \frac{2}{\rho}(x_l - z_j) \right)$ is supported on a ball of radius $\frac{\rho}{2}$ centered at $z_j$ we get $Q_{d+1}(\psi_j) = (\frac{2}{\rho})^d Q_{d+1}(\psi)$. From equation (42) it can be seen that $h_l$ is a linear combination of $\psi_j$ and hence by definition $Q_l(f) = \max_{x \in \mathbb{B}^n} \sum |f^{(l)}(x)|$ and substituting $C(n,d) = 2^{d+1} Q_{d+1}(\psi)$ we have

$$Q_{d_l+1}(h_l) \leq \max_j (P_l(z_j)) \frac{C(n,d)}{\rho^{d_l+1}} ||f_l \circ W_l||^{2d_l} \tag{43}$$

$$sup(Q_{d_l+1}(h_l)) \leq \max_j (P_l(z_j)) \frac{C(n,d)}{\rho^{d_l+1}} ||f_l \circ W_l||^{2d_l} \tag{44}$$

From equation (44) it is evident that

$$\mathcal{RC}_d(Z) \leq \frac{C(n,d)}{\rho^{d_l+1}} \frac{||f_l \circ W_l||^{2d_l}}{R_{d_l}(Z)} \tag{45}$$

From equation (24) in theorem (3) we know that

$$\frac{C(n,d)}{\rho^{d_l+1}} \frac{||f_l \circ W_l||^{2d_l}}{R_{d_l}(Z)} \geq (d_l+1)! \left( \frac{8n}{k\rho} \right)^{-d_l} \tag{46}$$

Simplifying equation (46) we bound $R_{d_l}(Z)$ as

$$\begin{aligned} R_{d_l}(Z) &\leq \left( \frac{8n}{k\rho} \right)^{d_l} \frac{C(n,d)}{\rho^{d_l+1}} ||f_l \circ W_l||^{2d_l} \\ &\leq \frac{C(n,d)}{\rho^{d_l+1}} \left( \frac{8n||f_l \circ W_l||^2}{k\rho} \right)^{d_l} \end{aligned} \tag{47}$$

Since $||f_l \circ W_l||^2 \leq ||f_l||^2 ||W_l||^2$ and as $||f_l||$ depends on the activation function which is a constant $||f_l \circ W_l||^2 \sim \mathcal{O}(||W_l||^2)$. The unknown constant $C(n,d)$ which arises in equation (47) has an effect on the final generalization bound and to capture its effect in all the layers we introduce a single unknown parameter $\gamma$ where $0 < \gamma \leq 1$. By the mathematical description of neural networks in equation (1), and using equation (10) which validates the multiplicative convergence (Kolmogorov; Bartle, 2014)

$$R_d(Z) = \frac{1}{\gamma}\left[\prod_{i=1}^{l}\left(\frac{8n}{k\rho}||W_i||^2\right)^{d_i}\right] \tag{48}$$

As we have assumed that the data to comes from a data distribution $p$ with finite mean and variance we can now apply the strong law of large to obtain the final expression

$$\mathbb{E}_{\mathbb{B}^n}\left[l(\mathcal{N})\right] \leq \mathbb{E}_Z[l(\mathcal{N})] + \mathcal{O}\left[\frac{1}{\gamma}\prod_{i=1}^{l}\left(\frac{8n}{k\rho}||W_i||^2\right)^{d_i}\right] \tag{49}$$

$\square$

## 5 COMPARISON WITH OTHER BOUNDS

Some of the most promising norm based methods like that of Neyshabur et al. use spectral norm of the matrix to give tight generalization bounds. However, such bounds have an issue of scaling abnormally high with the size of weight matrices (Vershynin, 2011) which showed that for an $m \times n$ random matrix $A$ with i.i.d entries the spectral norm is of order $\sqrt{m} + \sqrt{n}$, i.e. $||A||_2 \sim \mathcal{O}(\sqrt{m} + \sqrt{n})$ which scales with the number of nodes. Figure (1) demonstrates this on neural networks with varying depths trained on the MNIST dataset (LeCun & Cortes, 2010), the details of which has been relegated to the appendix.

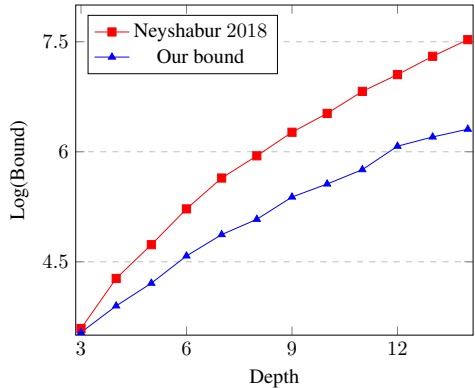

Figure 1: The comparison of our bound with spectrally-normalized margin bound (Neyshabur et al., 2018) with neural networks of increasing depths trained on the MNIST dataset.

## 6 CONCLUSION & FUTURE WORK

In this paper we introduced a novel generalization bound using geometric functional analysis. We have compared our approach to that of the previously existing spectral norm based generalization bounds and have shown the advantage of our method. We hope that this paper attracts more attention towards the use of geometric functional analysis in theoretical analysis of deep neural networks.

Future work includes extending this idea to specialized architectures such that various different properties of polynomials can be effectively leveraged to obtain architectural specific generalization bounds. Also finding an optimal value of the total number of training samples $k$ which minimizes the generalization error would be an viable direction to pursue.

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
