# OpenReview forum: "Improved Generalization Bound for Deep Neural Networks Using Geometric Functional Analysis"
_ICLR.cc/2022/Conference — ICLR 2022 Submitted_

### Official Review · Reviewer_SWDb · 2021-10-21

**Correctness:** 4
**Technical Novelty And Significance:** 2
**Empirical Novelty And Significance:** 2
**Recommendation:** 5
**Confidence:** 4

**Main Review:**

This paper is clearly written and well organized. I find it easy to follow. Also, this paper is technically sound. However, I think the assumption that activation functions of neural networks are polynomials with certain degrees, makes the contribution of this paper limited. Although many functions can be approximated by polynomials, this does not mean that replacing activation functions with polynomials is a good idea, since the errors can be accumulated layer by layer. Thus, the function assumed by authors could be far away from the actual function of a given neural network. Hope that the authors make more discussions on this issue.

**Summary Of The Paper:**

The authors study the generalization bound of neural networks. First, They assume that the activation functions as polynomials with certain degrees. Under this assumption, they prove a novel generalization bound using geometric functional analysis. The authors also compared their bounds with the method in (Neyshabur et al., 2018), showing that their bounds are better.

**Summary Of The Review:**

Although this paper is technically sound, I think the polynomial assumption on activation functions of neural networks is not very convincing, making the contribution of this paper limited.

---

> ### Author Response · Authors · 2021-11-23
> **Addressing the accumulation of error concern**
>
> We thank the reviewer for reviewing the paper , while we concede with the reviewer regarding the accumulation of the error , it is worth noting that the final expression involves only the degree and not the coefficients where the error is accumulated , once we fix the degree the final expression depends only on the initial degree used , so once proper approximation is made initially , it will not propagate through layers.

---

### Official Review · Reviewer_3Prr · 2021-10-26

**Correctness:** 3
**Technical Novelty And Significance:** 2
**Empirical Novelty And Significance:** 2
**Recommendation:** 3
**Confidence:** 4

**Details Of Ethics Concerns:**

No.

**Main Review:**

1. Maybe I lose some details, I am still puzzled on how to go from Eqn48 to Eqn49. Could the authors provide more details? Besides, could the authors explain how to go from sup over the Z (in Eqn 16) to mean over the Z (conclusion in Theorem5).

2. Could the authors explain the statement: "Another by-product of the analysis is that the bound obtained is in terms of the
Frobenius norm of the weight matrices rather than its spectral norm, which has shown to scale more rapidly with the size of the matrix". In my opinion, F-norm will produce a worse bound than the spectral norm.

3. I believe this paper can benefit a lot from more careful writing and some more precise statements. For example, could the authors validate the statement “this is the most suited framework for dealing with deep neural networks” (in the introduction part)?

4. This paper is not well organized. For example, Eqn3 and Eqn4 contain little information. Besides, some common techniques like the covering number are not encouraged to be introduced in the main text (since they are not strongly correlated to the main theorem). And each subsection in section3 is too short.

Some other comments:

1. In the second line of Eqn12, the symbol “\leq” may be improper.

2. There are some typos in the main text, for example, the introduction part: “it is widely in the ***”. Section3.5, f^l or f^{(l)}.
Missing ) in Eqn39.

3. What is the “strong law of large”, missing number?

4. Missing | in Eqn16.


**Summary Of The Paper:**

This paper aims at proving generalization bound via geometric functional analysis. Although the topic is interesting, this paper suffers from an unclear organization and there are some unclear points in the proof (see the following for more details). Therefore, I tend to give a "reject" and I hope the authors can provide more explanations for the theorems and carefully polish the writing in the next version.

**Summary Of The Review:**

It seems this paper is not in its shape to be published due to the above concerns, including the concerns on the proof and the unclear statements. More explanations on the theorem and the proof procedure may help reshape the paper.

---

> ### Author Response · Authors · 2021-11-23
> **addressing the concerns**
>
> 1).Using equation 16 in the pdf . it says that R_{d}(Z) measures the error on the bigger set compared to the smaller set , so the error being a polynomial as shown in the paper we can apply it here directly to the loss function although deterministically , to get eqn(49) from eqn(48).
>
> 2).We are sorry for the abuse of statements , we meant to state that the spectral norm is computationally hard to compute with limited resources , the frobenius norm being easy and also better in terms of numerical value of the bound as seen from experiments , it is possible that we attributed the improvement in performance to frobenius norm wherein some other forms in the expressions are responsible.
>
> 3 & 4) We will take proper care to rewrite the paper accommodating the suggestions given with reference to the writing style pointed out.

---

### Official Review · Reviewer_YPEk · 2021-10-30

**Correctness:** 1
**Technical Novelty And Significance:** 2
**Empirical Novelty And Significance:** 2
**Recommendation:** 1
**Confidence:** 4

**Main Review:**

I recommend rejection, for the reason that the theory in this paper (i.e. the main contribution) is either incorrect or missing many crucial steps. The statements and proofs brush many details under the rug and hence are very difficult to verify. Even if there is a correct underlying argument, I believe the paper would benefit from substantial revision and submission to a later venue.

Here are some of the concrete mathematical issues I found with the paper:

Initial theorems
- The definition of the Remez d-span R_d(Z) seems incorrect; it should be a supremum over all degree-d polynomials (the notation does not have a dependence on P, and e.g. Theorem 3 would not make sense if R_d(Z) is defined for a specific polynomial)
- Given this, Theorem 1 does not make sense. It seems to be saying that if there exists a polynomial P satisfying \mu_n(V_1(P)) >= \omega_d(Z) then there is an upper bound on R_d(Z). However, since R_d(Z) is a supremum over polynomials, it does not seem possible to upper bound it just by looking at P. Indeed, the proof seems to bound |sup_B P| / |sup_Z P| via Brudnyi-Ganzburg, but this does not bound R_d(Z).

Theorem 5
- In equation (43), why is ||f_l \circ W_l|| raised to the power (2d_l)?
- I don't understand how equation (44) implies equation (45). Certainly by definition RC_d(Z) <= sup Q_{d_l+1}(h_l) / sup Q_0(h_l), but now you seem to upper bound |max_j P_l(z_j)| / |sup Q_0(h_l)| <= |sup_Z h_l| / |sup_B h_l| <= 1/R_{d_l}(Z), but the last inequality goes in the wrong direction.
- I also don't understand how equation (46) is obtained from (45) and (24).
- The last sentence on page 8 is crucial to the whole proof, and does not make sense to me. Given upper bounds on R_{d_l}(Z), how do we upper bound R_d(Z)? Certainly it's not multiplicative. Equation (10) is about multiplicativity of degree when composing polynomials, but I don't see how that's relevant.
- What is the loss function used in the statement of Theorem 5?
- Why does 1/gamma disappear in the statement of Theorem 5? It's not a constant; it's exponential in the dimension.

Experiment
- Was the "constant" 1/gamma taken into account in experimentally computing the generalization bound?
- The closest-pair distance of the dataset was estimated by taking a subsample. This seems wrong; if there is a single pair of very close images, then the subsample closest-pair distance may be much larger than the true distance, which would artificially strengthen your generalization bound.

**Summary Of The Paper:**

This paper claims to contribute a novel generalization bound for deep neural networks based on treating the network as a polynomial (for polynomial activation function) with degree bounded by O(depth). This bound is experimentally compared against the norm-based bound from the prior work [NBS18].

**Summary Of The Review:**

I recommend rejection, for the reason that the theory in this paper (i.e. the main contribution) is either incorrect or missing many crucial steps. The statements and proofs brush many details under the rug and hence are very difficult to verify. Even if there is a correct underlying argument, I believe the paper would benefit from substantial revision and submission to a later venue.

---

### Official Review · Reviewer_nLh5 · 2021-11-03

**Correctness:** 4
**Technical Novelty And Significance:** 4
**Empirical Novelty And Significance:** 2
**Recommendation:** 6
**Confidence:** 3

**Main Review:**

**Strength**

This is a novel framework for the study of the generalization behavior of neural networks. Novel perspectives of the paper in my opinion are using a polynomial to approximate the activation function of neural nets and importing Vitushkin’s inequality to bound the generalization error via covering numbers. The tools introduced are likely to be useful for other analyses in a different context.

**Weakness**

* Approximating the activation function using a polynomial p(x) is unlikely to be accurate for unbounded x.
* The presented bound is algorithm-independent and is expected to be very loose for neural networks trained with SGD.
* More background literature on the generalization bounds of neural nets can be included. For example, "Information-Theoretic Generalization Bounds for Stochastic Gradient Descent" (Neu, 2021)
* The experimental verification seems lacking. The authors are encouraged to compare the developed bound with the true generalization error for various network architectures and datasets and compare the bound against other bounds developed for SGD.
* A question: In Figure 1, what is the degree of the polynomial approximating the activation function? How is the choice justified?


**Summary Of The Paper:**

The paper presents a new generalization bound for neural networks, which is shown to be tighter than the bound in [Neyshabur 2018].

**Summary Of The Review:**

The paper presents a novel framework for bounding the generalization error of neural networks. The techniques introduced in the paper are likely to have a broad impact. But the algorithm-independent nature of the developed bounds is likely to make the bound compare inferiorly to bounds derived specifically for SGD.

---

> ### Author Response · Authors · 2021-11-23
> **Addressing the weakness pointed out**
>
> Firstly , we  are thankful to the reviewer for carefully  going through the paper regarding the approximation part , it is true that for unbounded input approximation might not hold true , however in reality when dealing with practical data sets the data points are bounded and finite so we can assume that all data points come from within the sphere whose radius is the norm of the farthest point and the degree taken to approximate in the initial layer was taken to be 4 for better accuracy.
>
>
> In addition , the relevant background mentioned could very well be incorporated , nevertheless our attempt was to introduce a novel framework and build a theoretical perspective which helps understand neural networks better , definitely extending it to architecture specific networks would be our future endeavour.

---

### Official Review · Reviewer_FW5M · 2021-11-03

**Correctness:** 2
**Technical Novelty And Significance:** 2
**Empirical Novelty And Significance:** Not applicable
**Recommendation:** 3
**Confidence:** 3

**Main Review:**

Strengths:

The main strength of the paper is in bringing tools from geometric function analysis to understand generalization of neural networks. There are several tools which are used which seem interesting, and could potentially be valuable in deep learning theory.

Weaknesses:

Unfortunately there are several concerns regarding the presentation and the writing due to which I recommend rejection. To begin, it does not appear to me that the bound in Thm 5 actually has the nice properties that the authors claim. First, the authors say that in contrast to prior work their bound does not depend on spectral norm. However, I think the $\rho$ in Thm 5 could implicitly hide the spectral norm dependence. $\rho$ is the maximum value the polynomial can take in the domain, which just seems like spectral norm. Additionally, the authors say that the spectral norm guarantee is worse because for a mxn random matrix the spectral norm is $\sqrt{m}+\sqrt{n}$ which depends on $m$, but the Frobenious norm for such a matrix will scale with with $\sqrt{mn}$, which is much worse. Thm 5. additionally hides dependence on other parameters too. The big-oh notation hides $1/\gamma$, which is $C(n,d)$ which seems like it could be exponentially large in $n$ and $d$.

This brings me to a major concern regarding the presentation of the paper. The paper is not well-written, to the extent that it is difficult for me to do a sanity check of the results to make sure of their validity. To begin, the notation is not well-specified and is quite confusing. For instance, Thm. 5 uses $l$ both for the loss function and the depth of the network. The paper should clearly state in the beginning (and in the thereom too) what $n, d$ and $k$ are (in fact, I'm still not fully sure what $k$ is in the context of the neural network, not the polynomial). The proof also doesn't seem rigorous and I cannot be sure that it is correct, for e.g. the authors say in Eq. 49 that they use the law of large numbers to get the concentration, but what is needed here is a finite sample guarantee, not asymptotic convergence which the strong law of large numbers gives. The right concentration bounds need to used, and the failure probability needs to be analyzed here. In fact, the generalization bound in Thm 5 does not even depend on the number of samples, which is strange.

The authors should also spend much more space discussing the tools from geometric function analysis. Some of the proofs can be moved to the appendix, and the entire analysis in page 3 for products of polynomials seems a bit trivial and unnecessary to spend so much time on, when there's more interesting stuff to talk about.

The paper also seems to say that it is the first to derive covering number bounds for neural networks, but these were also derived by Bartlett-Foster-Telgarsky.

The idea of using geometric function analysis for deep learning theory seems interesting and I encourage the authors to pursue this more, but the paper falls short of the bar.

Typos:

1. page 1: 'it generalization problem'
2. page 4: 'lower on the variation of a function'
3. it should be mentioned that the norm is the Frobenious norm in Thm 5.

**Summary Of The Paper:**

The paper considers the problem of obtaining generalization bounds for deep neural networks. The paper uses tools from geometric function analysis to derive bounds for the covering number of neural networks, and derive generalization bounds using the covering numbers. The main strength of the bound as claimed by the paper is that they depend on Frobenius norm rather than the spectral norm, but I have several concerns about the results.

**Summary Of The Review:**

Due to the above issues regarding the results and the presentation, I recommend rejection for the paper.

---

### Decision · Program_Chairs · 2022-01-20

**Decision:**

Reject

**Comment:**

The paper provides a new geometric functional analysis perspective for the generalization bounds for neural networks. As the AC, I actually quite liked the twist the authors are providing for this particular work. Unfortunately, the current presentation is too crude to provide an elementary picture for the developments and I strongly encourage the authors to revise the paper for the next deadline based on the remarks from the reviewers.